# Modification of Sympathetic and Hypothalamic Responses to Prevent Complications of COVID-19: "Dam and Wall Concept"

**Sanjiv K. Hyoju** [1,2] 🆔

1  Bhomi Hospital and Medical Research Institute, Banepa 45210, Nepal; skhyoju@gmail.com;
   Tel.: +1-7184905674
2  Surgical Infection and Therapeutic Laboratory, Biological Science Division, University of Chicago,
   Chicago, IL 60637, USA

**Abstract:** We are in the midst of the COVID-19 pandemic. Since December 2019, severe acute respiratory coronavirus (SARS-CoV-2) has infected more than half a billion people, killing nearly 7 million people worldwide. Now various variants of SARS-CoV-2 are causing mayhem and driving the global surge. Epidemiologists are aware of the fact that this virus is capable of escaping immunity and likely to infect the same person multiple times despite adequate vaccination status. Elderly people and those with underlying health conditions who are considered high-risk are likely to suffer complications. While it is tempting to frame complications and mortality from COVID-19 as a simple matter of too much of a virulent virus in too weak of a host, much more is at play here. Framing the pathophysiology of COVID-19 in the context of the Chrousos and Gold model of the stress response system can shed insight into its complex pathogenesis. Understanding the mechanisms of pharmacologic modification of the sympathetic and hypothalamic response system via administration of clonidine and/or dexamethasone may offer an explanation as to why a viral pathogen can be well tolerated and cleared by one host while inflaming and killing another.

**Keywords:** stress response system; sympathetic activity; HPA (hypothalamic–pituitary–adrenal) axis; SARS-CoV-2; catecholamine; corticosteroids; clonidine; dexamethasone

## 1. The Role of Sympathetic Nervous System and Hypothalamic–Pituitary–Adrenal (HPA) Axes That Drive Homeostasis in the Stress Response System

Body homeostasis is defined as a complex dynamic yet balanced physical and biological status maintained by all living creatures for survival. The term was first coined by Walter Cannon [1]. Body homeostasis leading to uneventful recovery can become easily disrupted in the face of various intrinsic and extrinsic stressors such as bacterial and viral infections and environmental factors such as physical or psychological trauma. As a result, maladaptive responses can occur with sympathetic hyperactivity leading to neurohormonal immune activation that may continue inexorably even after absence of the perturbing forces. This complex brain–body response is known as the stress system response, as first described by Chrousos and Gold [2,3]. In this model, two axes are primarily described, (A) a sympathetic nervous system (SNS) outflow axis and (B) the hypothalamic–pituitary–adrenal (HPA) axis. During stress, homeostasis is achieved via fine-tuned control of the SNS and HPA axes so that they remain tightly coordinated to produce an appropriately measured response toward host recovery (Figure 1A).

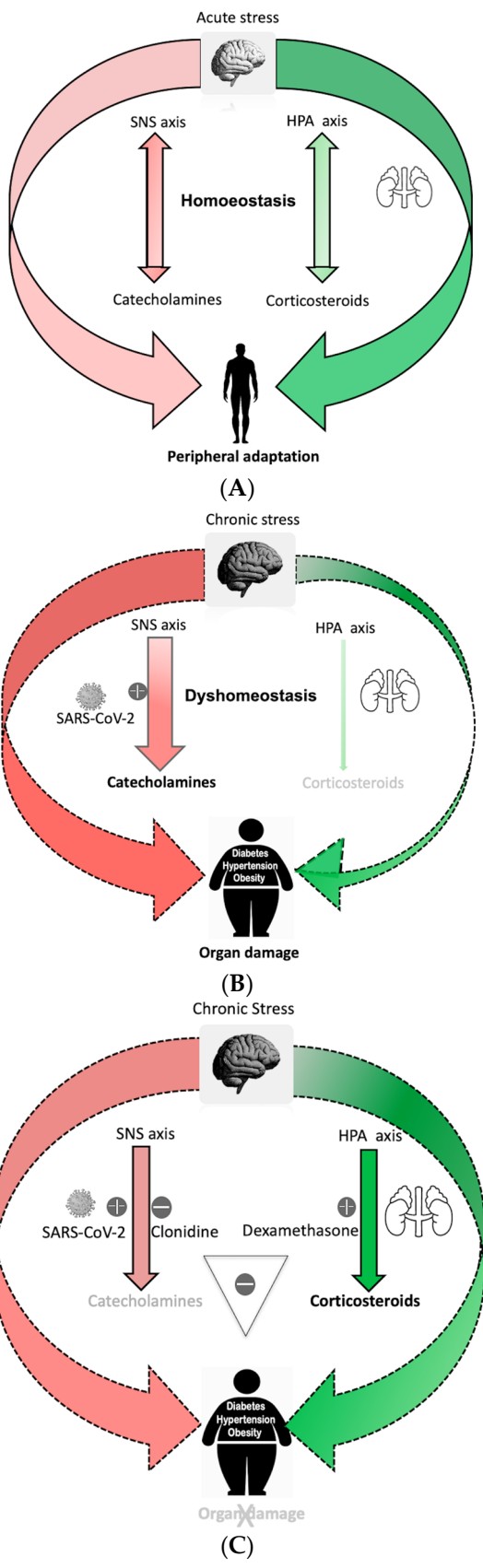

**Figure 1.** (**A**). Processing of acute stress: Acute stress activates the sympathetic nervous system (SNS) axis and the hypothalamic–pituitary–adrenal (HPA) axis to maintain homeostasis. Repetitive acute

stress leads to peripheral adaptation. (**B**). Imbalanced stress response system during chronic stress with gain of function of SNS axis and loss of function of HPA axis, cause dyshomeostasis. SARS-CoV-2 over-activates SNS axis, causing a pro-inflammatory catecholamines surge leading to organ damage in the susceptible person. (**C**). Clonidine blocks SNS axis and dexamethasone enhances the HPA axis, acting synergistically to balance the stress response system during SARS-CoV-2 infection and prevent organ damage in a dyshomeostatic person.

The SNS axis is mediated by catecholamines. This system's input and output signals drive the interaction between the brain and the immune system. During acute stress, catecholamines surge in the body to prepare for a "fight and flight response", queueing up various behavior (increased arousal, alertness, loss of sleep and appetite) and physiological changes (increased heart rate and blood pressure). As a result, immunomodulatory pathways are activated via various adrenoreceptors (ARs) present on immune cells. Evidence for this coordinated response can be observed by sympathoadrenergic nerve fibers that are abundantly present on the immune cells that respond to catecholamines released during stress [4]. Yet once this system is activated at the whole-organ level, it can be both beneficial or deleterious depending on its intensity, duration and whether prior pre-conditioning of immune cells has occurred. Additionally, the density and affinity of ARs and the concentration of norepinephrine in local organs can differentially express the intensity of the immune response. For example, it has been shown that norepinephrine has preferentially stronger affinity for $\alpha$ARs 1 and 2 on immune cells, resulting in a predominantly pro-inflammatory response. In contrast, a high concentration of norepinephrine activates $\beta$ARs [5–7]. $\beta_2$ adrenoreceptor activation inhibits the production of pro-inflammatory cytokines such as IL-12, TNF$\alpha$ and interferon gamma while also stimulating the production of anti-inflammatory cytokine IL-10 [8]. Hence, depending upon the type of AR population activated on immune cells, the immunomodulatory response might be either pro- or anti-inflammatory. Similar to the SNS axis, the HPA axis plays an equally important role in maintaining homeostasis following stress-related perturbations. The HPA axis increases peripheral levels of glucocorticoids (GCs). Glucocorticoids bind to intracellular glucocorticoid receptors (GRs) in peripheral immune cells and translocate to the nucleus; this downregulates NF$\kappa$B pro-inflammatory gene transcription that can encode various cytokines such as IL-6, IL-1 and TNF-$\alpha$ [9]. Thus, peripherally released glucocorticoids have a primarily anti-inflammatory effect. Likewise, both SNS and HPA axis play major roles in redistributions of T cells. An increase in plasma cortisol reduces the blood lymphocyte count, whereas catecholamines generally cause leukocytosis. Both CD8+ T cells and natural killer cells rapidly yet transiently increase in the blood following catecholamine infusion, which can be mitigated by catecholamine inhibition [10].

As a fight-or-flight response, sympathetic SNS axis activation not only prepares the body physically to manage stress, but it also induces a pro-inflammatory response that can decrease and shift to an anti-inflammatory response. Activation of the SNS axis aims to localize the inflammatory response and protect the body from any detrimental effects of released pro-inflammatory cytokines [11]. Concomitant activation of an anti-inflammatory HPA axis further shuts down ongoing inflammation in an effort to maintain homeostasis by preventing excessive collateral damage to organs. Thus, there is coordination of both a central and peripheral stress response that adaptively interact to mitigate stress (Figure 1A).

## 2. Chronic Stress Disrupts the Stress Response, Resulting in Maladaptation and Impaired Recovery

Selye et al. [12] defined the physiologic stress response as a biological phenomenon that seeks to balance host defense against the stressor while limiting internal damage. Repetitive and continuous stress eventually results in a maladaptive response to a harmful stimulus. Allostasis, the process by which the body responds to stressors in order to regain homeostasis [13], includes recalibration of the SNS axis and HPA axis to realign

immunological functions of the body toward recovery. An excessive "allostatic load" can result in organ damage [14] despite the body continuing to calibrate itself to the ongoing stress [15]. Prolonged duration of the allostatic load can lead to complete failure of the system to recalibrate itself. In some cases, the SNS and HPA axes may develop a new set point [16] to minimize collateral damage. This causes an imbalance of the central stress response system by affecting the calibrating efficiencies of these axes. Goldstein et al. refered to this condition as dyshomeostasis [3]. The predominant outcome in dyshomeostasis is overactivity of the SNS axis perturbating immune cell function locally due to continuous release of norepinephrine (NE), leading to perhaps a "gain of function", i.e., hypersensitization with increased response to the same level of stimulation, primarily leading to a pro-inflammatory response. Such gain of function can augment the production of macrophage derived TNF$\alpha$ through $\alpha_2$ARs [17], drive CD8+ T lymphocytes toward a more pro-inflammatory phenotype, and activate more $\beta_2$ adrenoreceptors on the immune cell, producing a pro-inflammatory response instead of the usual anti-inflammatory response [18,19]. Another outcome is HPA axis overactivity leading to decreased sensitivity of glucocorticoid receptors to glucocorticoids. This perhaps could be a "loss of function", i.e., desensitization with decreased response to the same level of stimulation, reducing its anti-inflammatory effects. Such biological changes are attributed to the pro-inflammatory phenotype [20]. Because of the altered phenotype of immune cells during dyshomeostasis and superimposed stress due to virus infection [21], NE loses its ability to localize the inflammation and fails to protect the host from the detrimental effect of cytokines. Perhaps this response might be erratic and detrimental to health [22].

## 3. Body Dyshomeostasis and Role of Sympathetic Hyperactivity

The current literature supports the notion that chronic health diseases such as obesity, hypertension, diabetes, autoimmune diseases and cardiovascular disease are linked to chronic sympathetic hyperactivity and hence a chronic proinflammatory response [23]. The rapid evolution of civilization and its accompanying changes in living style, psychological stress, diet such as high-fat low-fiber diet and resultant dysbiosis [24–27] are major contributors to the chronic disease state. As such, high-risk populations are likely to be disproportionately impacted following any form of additional stress. A model developed in the Alverdy lab captures many of the features of this so called "dyshomeostasis" state. In this model, healthy mice consuming high-fiber low-fat diet (chow diet) all survived after being subjected to a major operative stress of partial hepatectomy, whereas a 60–70% mortality rate was observed among similarly treated obese mice consuming a high-fat low-fiber diet. Both groups of mice were starved overnight, received antibiotics prior to operation and underwent surgery under strict aseptic precaution. Yet the consumption of a high-fat low-fiber diet led to such a dramatic alteration in outcome in this model such as dysbiosis and the emergence of a gut pathobiome that caused marked endogenously derived stress to the mice [28]. Intriguingly, this model may represent a state of dyshomeostasis-induced obesity whereby there is chronic sympathetic hyperactive inflammation that contributes to organ failure and death during additional stress (Figure 1B).

## 4. SARS-CoV2 Imbalances Stress Response System in Vulnerable Population

It is noteworthy to acknowledge that 80% of patients who develop a SARS-CoV-2 infection have mild symptoms. These are mainly young and healthy individuals with presumably a balanced stress response system. Elder age group and those living in a state of dyshomeostasis with underlying medical comorbidities are at risk of developing severe COVID-19 symptoms and are more predisposed to die from organ failure (Figure 1B). It has now been well established that COVID-19 cause autonomic nervous system dysfunction in human beings [29]. This is a neurotropic virus that is known to reach the brainstem directly or indirectly, leading to impaired autonomic function with increasing SNS axis hyperactivity [22,30–32]. The pathogenesis of COVID-19 reveals a significant role of a sympathetic-hyperactivity-mediated imbalance in angiotensin converting enzyme 1 (ACE1)

Vs ACE2 in the evolution of its disease sequalae and mortality [31,33–35]. One of the leading causes of death in COVID-19 is hypoxia from acute respiratory distress syndrome secondary to viral pneumonia. One possibility to explain the differential response of young healthy versus elder infirm patients' outcomes following COVID-19 infection could be excessive reactive malfunction of the autonomic nervous system with sympathetic hyperactivity and hyperinflammation [36]. For example, a stellate ganglion blockade with local anesthesia to interrupt sympathetic outflow to the lungs has been proposed as an intervention to prevent acute respiratory distress syndrome [37]. Animal studies have demonstrated attenuation of acute lung injury following this approach [38]. Similarly, human long COVID-19 symptoms have been significantly improved after a similar intervention [39]. These observations may indicate a central theme across these disease states as applied to COVID-19 that the stress response systems with SNS axis overactivation are a main driver for the immunopathology observed in COVID-19 pneumonia [31,33]. Hypercoagulability, myocardial infarction, thrombosis and stroke are another spectrum of severe COVID-19 sequalae leading to morbidity and mortality. Biological markers in COVID-19, such as low-platelet-count, deranged PT, PTT, protein C level and elevated D-dimer indicating coagulopathy, have been associated with increased circulating catecholamines [40,41]. Similarly, an increased incidence of Takasubo cardiomyopathy in COVID-19 patients has been linked to cytokine storm and sympathetic-hyperactivity-related stress [42]. It is well known that increased catecholamines induce the release of IL-6 and TNFα cytokines, causing leukopenia and orchestrating immune dysregulation, perpetuating cytokine storm through a self-amplifying loop within macrophages [43]. Such phenomena have been observed in COVID-19 patients. Disturbances of HPA and SNS axes responses have been implicated for the increase in C reactive protein (CRP), IL-6 and the incidence of leukopenia in the setting of metabolic syndrome with chronic disease [44]. Derangement of such biomarkers in chronic disease patients after being infected with SARS-CoV-2 [45] is suggestive of the exacerbation of disturbances in the SNS and HPA axes responses and thus may correlate with poor outcome [45,46].

## 5. Clonidine and Dexamethasone Act Synergistically to Prevent Complication during SARS-CoV-2 Infection

We have previously hypothesized that SARS-CoV-2 infection leads to overactivation of the SNS axis and drives uncontrolled inflammation in the chronic sympathetic hyperactive population leading to poor outcome. Pharmacologic attenuation of SNS overactivation can be addressed by an FDA approved agent, clonidine, an alpha-2 agonist that may have clinical benefit and prevent COVID-19 complications [22,47]. In a small case series, we demonstrated early administration of clonidine-mitigated SARS-CoV-2 related symptoms, thus preventing complications [47]. Others have used clonidine lately in CCU as a sedation and ventilation method to manage respiratory distress in SARS-CoV-2 patients [48]. Similarly, retrospective analysis performed by Hamilton et al. demonstrated that early used of an alpha-2 agonist is associated with reduced 28-day mortality and later use of the medication is not effective [49]. Baller et al. recommended the use of clonidine as prophylaxis against delirium in SARS-CoV-2 patients [50].

Counterintuitively, pharmacologic enhancement of the HPA axis with corticosteroid treatment has been found to be of clinical benefit in COVID-19 patients; however, the timing of administration seems to play an important role [51]. Multiple clinical trials have tested the effectiveness of dexamethasone in COVID-19 patients. The RECOVERY trial provided evidence that treatment with dexamethasone is beneficial for COVID-19 patients who required oxygen support, although it was not helpful for those patients who did not require oxygen [52]. Although the CoDex randomized clinical trial demonstrated significant increase in the number of ventilator-free days over 28 days with dexamethasone treatment in moderate to severe COVID-19 patients, there was no difference in the mortality rate. Early treatment with dexamethasone has no added benefit in SARS-CoV-2 infection outcome and could perhaps possibly harm [53] patients due to secondary infection [54].

Recently, it was revealed that SARS-CoV-2 targets the adrenal gland and can cause adrenal insufficiency [55].

Taken together, much evidence suggests that disruption of the stress response system with the gain of function of the SNS axis and loss of function of the HPA axis can lead to a worse outcome during SARS-CoV-2 infection. Therefore, here I propose that timely administration of a combination of clonidine and dexamethasone in high-risk patients has the potential to prevent complications and death (Figure 1C).

Clonidine should be given in symptomatic SARS-CoV-2 infection and should be started early in high-risk population and increased gradually while monitoring blood pressure and heart rate. If the patient's condition has not improved or the patient requires oxygen and/or has deranged biomarkers (such as serum CRP, D-dimer, serum ferritin), dexamethasone should be added without delay at a tolerable dosage (Figure 2).

**Clonidine +/- Dexamethasone treatment in symptomatic SARS-CoV-2 positive patients**

**Clonidine initiation**

**\*\*Check blood pressure (BP) and heart rate (HR)**

| BP, mmHg | 100-119/60-79 | 120-139/80-89 | ≥140/90 |
|---|---|---|---|
| Clonidine, µg/day | 100 | 200 | 300 |

**Clonidine escalation and maintenance**
If symptoms persist or don't improve by 24 hours, increase dose of clonidine by 100µg every 24 hours till symptoms improve or reach maximum dose of 600µg/day
Continue same dose of clonidine for 48 hours once symptoms improve

**Clonidine tapering**
Decrease dose of clonidine by 100-200µg every 24 hours
e.g. : 600µg>>400µg>>300µg>>200µg>>100µg>>stop

**Precaution**
• **\*\*Hold or don't start clonidine if BP <100/60mmHg or HR <60/min**
  ➤ Give 500ml-1000ml of fluids >> check BP & HR in one hour >> start or continue clonidine if BP ≥100/60mmHg and HR ≥60/min
  ➤ Offer plenty of fluids and avoid upright position for two hours after taking clonidine to avoid dizziness
• Hold or decrease other antihypertensive medications
• Hold other sedative medications

**Dexamethasone combination**
• Combine clonidine with dexamethasone if persistent fever for >7days, intense cough, hemoptysis, $SpO_2$<88% for >4hours or elevated serum biomarkers
• Start dexamethasone at 6mg-18mg/day and continue till symptoms improve or serum biomarkers trend down

**Figure 2.** Protocol design for initiation, maintenance and tapering of clonidine and dexamethasone in symptomatic SARS-CoV-2 positive patient.

Following is a case discussion to support the above proposal. A 54-year-old overweight female with past medical history of hypertension and depression tested positive for SARS-CoV-2 infection via nasopharyngeal swab PCR. She presented to the emergency department with a history of fever for 3 days, temperature maximum up to 39 °C, cough for 10 days and shortness of breath with excessive fatigue for 1 day during an acute surge of COVID-19 with delta variant. She was admitted to the COVID ICU and started on continuous positive airway pressure ventilation. She received IV dexamethasone 6 mg twice daily along with broad spectrum antibiotics, antifungal and heparin. Her condition deteriorated requiring 80% $FiO_2$ to maintain $SpO_2$ above 90%. Blood tests demonstrated elevated serum ferritin, D-dimer, CRP, PT and LDH (Table 1). A high-resolution CT scan of her chest demonstrated scattered areas of ground glass opacities and consolidation in bilateral lung fields with subcutaneous emphysema with CT severity score of 24/25 and CORAD score 6 (Figure 3). Her condition was not improved and she consulted me virtually on the 15th day of onset of symptoms. After taking informed consent, clonidine 100 microgram every 8 h was started on the same day and gradually increased up to 200 microgram every 6 h by day 20th of onset of symptoms monitoring her heart rate and blood pressure. Both hypoxia and tachypnea improved after 5 days of starting of clonidine. Clonidine was gradually tapered off and stopped over next 10 days. Other cases of moderate to severe COVID-19 patients treated with clonidine and/or dexamethasone during acute surge of cases in the Kathmandu and Banepa Valley from May 2020–September 2021 are outlined (Table 2).

**Table 1.** Lab investigation of patient: blood parameters. CRP: C reactive protein, PT: prothrombin time, LDH: Lactate dehydrogenase.

| Parameter | | Normal Range |
|---|---|---|
| Serum ferritin | 1006.4 | 11.0–306 ng/mL |
| D-dimer | 4.6 | <0.5 mg/L |
| CRP | 138.47 | <10 mg/dL |
| PT | 20.5 | 11–16 s |
| LDH | 926 | 0–246 ng/mL |

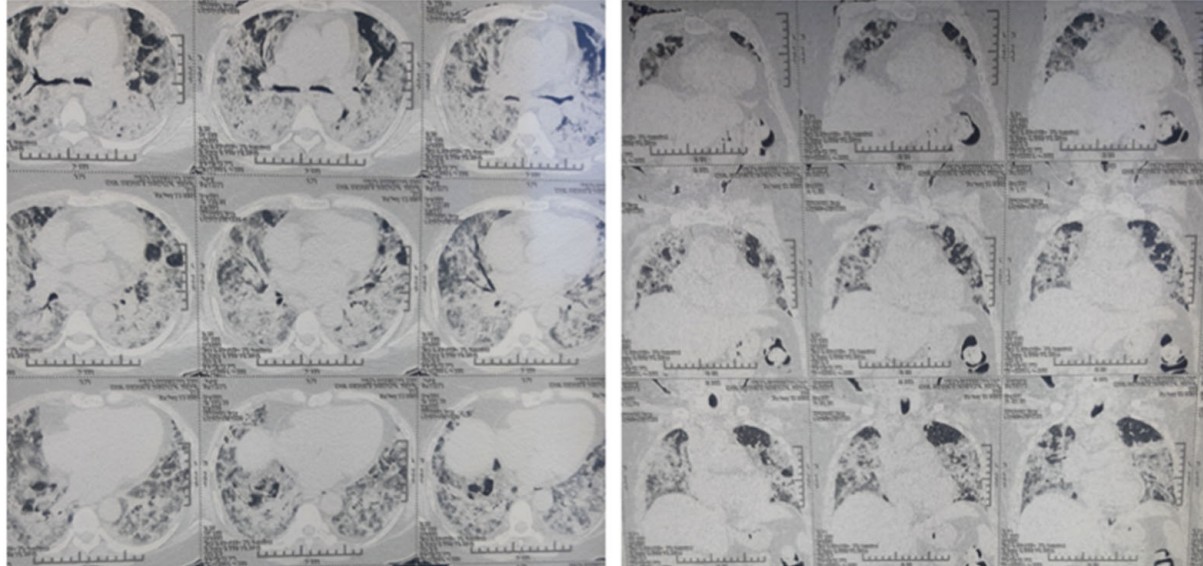

**Figure 3.** High-resolution CT scan of chest demonstrating scattered areas of ground glass opacities and consolidation in bilateral lung fields with subcutaneous emphysema with CT severity score of 24/25 and CORAD score 6.

**Table 2.** Patients treated during acute SARS-CoV-2 surge in Kathmandu and Banepa Valley from May 2020 till September 2021.

| Age/Sex | PCR Test Result | Symptoms/Lowest Recorded SpO$_2$/Oxygen Supplementation Received (Yes/No) | Abnormal Laboratory Reports (Normal Value) | Home/Hospital Based Treatment | Highest Doses of Clonidine and/or Dexamethasone Treatment per Day | Total Duration of Treatment with Clonidine | Outcome |
|---|---|---|---|---|---|---|---|
| 61 y/female | Unknown | High-grade fever, cough, shortness of breath/ SpO$_2$:85/Yes | ESR:64 mm/h (0–10) CRP: ++ D dimer: 1.61 µg/mL (<0.5) Ferritin: 721 ng/mL (6.24–264) AST: 138 U/L (5–40) ALT: 153 U/L (5–45) | Home-based treatment | Clonidine 300 microgram/day | 7 days | Recovered |
| 49 y/female | Positive | Fever, cough, hemoptysis, shortness of breath/SpO$_2$/90/Yes | D dimer 1.01 mg/L (<0.5) | Home-based treatment | Clonidine 300 microgram/day | 10 days | Recovered |
| 51 y/male | Positive | Fever, cough, loss of smell, loss of taste, weakness/ SpO$_2$:88/No | CRP: ++ ESR:16 mm/h (0–10) LDH: 486 U/L (<460) AST:124 U/L (5–40) ALT:101 U/L (5–45) CT Chest: CORADS-6 | Home-based treatment | Clonidine 300 microgram/day | 10 days | Recovered |
| 79 y/female | Positive | Fever, cough, shortness of breath/ SpO$_2$:85/No | TLC < 3500 (4000–11000) | Home-based treatment | Clonidine 300 microgram/day | 10 days | Recovered |
| 67 y/female | Positive | Fever, shortness of breath, weakness/ SpO$_2$: 88/Yes | ESR:52 mm/h (0–20) CRP:48 mg/L (0–6) Ferritin: 389 ng/mL (11–307) | Home and hospital-based treatment | Clonidine 300 microgram/day and dexamethasone 6 mg/day | 10 days | Recovered |
| 32 y/male | Positive | Fever, cough, shortness of breath/ SpO$_2$:75/Yes | CRP:31 mg/L (0–6) CT Chest: CORADS-6 LDH:400 U/L (120–246) | Hospital-based treatment | Clonidine 600 microgram/day and dexamethasone 12 mg/day | 30 days | Recovered |

**Table 2.** *Cont.*

| Age/Sex | PCR Test Result | Symptoms/Lowest Recorded SpO$_2$/Oxygen Supplementation Received (Yes/No) | Abnormal Laboratory Reports (Normal Value) | Home/Hospital Based Treatment | Highest Doses of Clonidine and/or Dexamethasone Treatment per Day | Total Duration of Treatment with Clonidine | Outcome |
|---|---|---|---|---|---|---|---|
| 48 y/female | Positive | High-grade fever, cough, shortness of breath, loss of smell, taste and appetite/SpO$_2$: 75/Yes | CRP = 59 mg/L (0–6) ESR: 53 mm/h (0–10) | Home-based treatment | Clonidine 600 microgram/day and dexamethasone 9 mg/day | 14 days | Recovered |
| 59 y/male | Positive | High-grade fever, cough, shortness of breath, loss of smell, taste and appetite/SpO$_2$: 75/Yes | ESR: 38 mm/h (0–10) CRP:42 mg/L (0–6) NT-Pro BNP: 20211 pg/mL (<220) | Home-based treatment | Clonidine 600 microgram/day and dexamethasone 12 mg/day | 21 days | Recovered |
| 43 y/male | Positive | High-grade fever, cough, shortness of breath, loss of smell, taste and appetite, SpO$_2$:80/Yes | ESR: 54 mm/h (0–10) CRP: 57 mg/L (0–6) NT-Pro BNP: 8567 pg/ml(<220) | Home-based treatment | Clonidine 600 microgram/day and dexamethasone 9 mg/day | 18 days | Recovered |
| 38 y/male | Positive | High-grade fever, cough, shortness of breath/SpO$_2$: 90/Yes | CRP: 211 mg/L (0–5). ESR: 83 mm/h. (0–12) Serum ferritin > 3000 ng/mL(25–350) LDH: 535 U/L (125–220) D dimer: 600 ng/mL (<500) AST: 361 U/L (16–63) ALT: 125 U/L (16–37) | Home-based treatment | Clonidine 300 microgram/day and dexamethasone 16 mg/day | 14 days | Recovered |
| 69 y/female | Positive | Fever, cough, shortness of breath/ SpO$_2$:85/Yes | ESR: 30 mm/h (0–9) CRP: 24 mg/L(<6) Procalcitonin 0.81 ng/mL (0–0.5) | Home-based treatment | Clonidine 300 microgram/day and dexamethasone 12 mg/day | 21 days | Recovered |

**Table 2.** *Cont.*

| Age/Sex | PCR Test Result | Symptoms/Lowest Recorded SpO$_2$/Oxygen Supplementation Received (Yes/No) | Abnormal Laboratory Reports (Normal Value) | Home/Hospital Based Treatment | Highest Doses of Clonidine and/or Dexamethasone Treatment per Day | Total Duration of Treatment with Clonidine | Outcome |
|---|---|---|---|---|---|---|---|
| 72 y/male | Positive | Fever, cough, shortness of breath/ SpO$_2$:80/Yes | CRP: 38 mg/L (0–10) CT Chest: CORADS-6 | Isolation-center-based treatment | Clonidine 300 microgram/day and dexamethasone 6 mg/day | 15 days | Recovered |
| 59 y/female | Positive | Fever, cough, shortness of breath, SpO$_2$:82,Yes | ESR: 33 mm/h (0–12) CRP: 19.8 mg/L (0–6) Glucose ®: 206 | Home-based treatment | Clonidine 300 microgram/day and dexamethasone 6 mg/day | 12 days | Recovered |
| 58 y/male | Positive | Fever, cough, whole body ache, shortness of breath, intense headache, BP:197/117 mmHg/ SpO$_2$:82/Yes | CRP: 89 mg/L (<10) D-Dimer: 1133 ng/mL (30–400) | Home-based treatment | Clonidine 900 microgram/day and dexamethasone 12 mg/day | 18 days | Recovered |
| 45 y/male | Positive | Fever, cough, hemoptysis/SpO$_2$: 88/No | CRP: 88 mg/L (0–6) Ferritin: 544 ng/mL (17–464) AST: 48 U/L (5–40) ALT: 86 U/L (5–45) | Home treatment | Clonidine 300 microgram/day and dexamethasone 12 mg/day | 14 days | Recovered |
| 96 y/male | Unknown | Fever, shortness of breath/SpO$_2$: 86/Yes | Not done | Home-based treatment | Clonidine 400 microgram/day and dexamethasone 6 mg/day | 10 days | Recovered |

Given the variable and potentially opposing effects of clonidine and dexamethasone, either drug alone may not be sufficient to control the ongoing inflammation. Randomized clinical trials to test the hypothesis that clonidine and dexamethasone can act synergistically to stop the ongoing overt inflammation centrally at the brainstem level and peripherally at the organs should be encouraged. Generation of the appropriate biomarkers and proper internal control would yield considerable mechanistic information useful to those unfortunate individuals who develop severe symptoms following COVID-19 infection. This combined approach recapitulates the idea of "closing dam (centrally) and building wall (peripherally) to protect the home field (major organ)" from excessive catecholaminergic flooding. It should be emphasized that late administration of either drug is not beneficial after significant organ damage has already occurred.

## 6. Concluding Remark

The COVID-19 pandemic remains a clear and present danger to mankind. Multiple variants are now emerging and hospitalizations are increasing. In the healthy population, repetitive infection with SARS-CoV-2 causes a flu-like phenomenon because of a balanced stress response system. For those with imbalanced stress response system, additional stress due to SARS-CoV-2 infection leads to substantial morbidity and mortality. Given the multiple ongoing emergences with new variants of SARS-CoV-2 virus, early recognition of symptomatic and high-risk patients whose response represents "dyshomeostasis" is critical and treatment with already available agents should be encouraged. Randomized control clinical trials on clonidine alone and combination of clonidine and dexamethasone in symptomatic SARS-CoV-2 infection as mentioned in the protocol (Figure 2) is necessary to test this challenging "Dam and Wall" concept to prevent any further complications by emerging new variants of SARS-CoV-2 virus.

**Funding:** This research received no external funding.

**Data Availability Statement:** Not applicable.

**Acknowledgments:** I would like to acknowledge Surichhya Bajracharya, Advocate Illinois Masonic Medical Center, for editing this manuscript.

**Conflicts of Interest:** The author declares no conflict of interest.

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
