# Peer review of "Modification of Sympathetic and Hypothalamic Responses to Prevent Complications of COVID-19: “Dam and Wall Concept”"

_stresses, doi:10.3390/stresses3010012_

Round 1

Reviewer 1 Report

This is an interesting review suggesting possible mechanisms whereby the interaction of Covid with the stress axis explains the susceptibility of some individuals to severe symptoms and death. Furthermore, suggested treatments and a single case report with treatment based on this are reported. This is an interesting perspective with far-reaching consequences given the prevalence of the infection and paucity of current understanding. The review is well written with good structure. I am not qualified to comment on the clinical aspects of the case report. My one comment is that I am surprised the author did not include potential direct effects on adrenal axis function (eg see https://www.nature.com/articles/s41574-022-00700-8) and at least some mention of this should be made.

Author Response

Thank you very much for your comment. I  discussed about the article that reviewer mentioned and updated the reference. 

Thank you 

Reviewer 2 Report

attached file

Author Response

Thank you very much for your extensive comments. Here are my responses. Please refer to attach document. 

Thank you 

Round 2

Reviewer 2 Report

Lm 38-39 : there are NO projections from the LC to the intermediolateral cell column (IML) and sympathetic system : please just say fine tuning of SNS. Delete LC/NE throughout the ms and figures.

All the projections to the IML and SNS come from the glutamate and adrenaline cell bodies in the rostral ventrolateral medulla, and possibly from the A5 noradrenergic cell group.

Lm 78: can decrescendo ? you mean decrease? Please change over.

Lm 100 : define the gain of function exactly as you do in your rebuttal to R2.

LM 141: and low baseline inflammation

Lm: 163: typo: infarction.

Lm 188: Hamilton is 49 if I am correct. Please check the lag in your references.

Thanks for the field job in Kathmandu valley and proposing a rational treatment, even if given conditions the trial is not randomized: the author should be commanded for carrying on this field work. This referee R2 and the author both know that we are not anywhere close to a demonstration. Please add one such sentence in conclusion : the lay reader should know that clonidine+dex may be useful but a randomized trial is needed.

Author Response

Dear Reviewer-2

Thank you for your more comments. Here is my response.

Lm 38-39 : there are NO projections from the LC to the intermediolateral cell column (IML) and sympathetic system : please just say fine tuning of SNS. Delete LC/NE throughout the ms and figures.

Done

All the projections to the IML and SNS come from the glutamate and adrenaline cell bodies in the rostral ventrolateral medulla, and possibly from the A5 noradrenergic cell group.

Lm 78: can decrescendo ? you mean decrease? Please change over.

Change to decrease 

Lm 100 : define the gain of function exactly as you do in your rebuttal to R2.

Exactly define 

LM 141: and low baseline inflammation

not able to identified. 

Lm: 163: typo: infarction.

Corrected

Lm 188: Hamilton is 49 if I am correct. Please check the lag in your references.

Checked 

Thank you